# Ionic Nanocomplexes of Hyaluronic Acid and Polyarginine to Form Solid Materials: A Green Methodology to Obtain Sponges with Biomedical Potential

**DOI:** 10.3390/nano9070944

**Published:** 2019-06-29

**Authors:** María Gabriela Villamizar-Sarmiento, Ignacio Moreno-Villoslada, Samuel Martínez, Annesi Giacaman, Victor Miranda, Alejandra Vidal, Sandra L. Orellana, Miguel Concha, Francisca Pavicic, Judit G. Lisoni, Lisette Leyton, Felipe A. Oyarzun-Ampuero

**Affiliations:** 1Advanced Center of Chronic Diseases (ACCDiS), Universidad de Chile, Santos Dumont 964, Independencia, Santiago 8380494, Chile; 2Departamento de Ciencias y Tecnología Farmacéuticas, Facultad de Ciencias Químicas y Farmacéuticas, Universidad de Chile, Santos Dumont 964, Independencia, Santiago 8380494, Chile; 3Instituto de Ciencias Químicas, Facultad de Ciencias, Universidad Austral de Chile, Isla Teja, Casilla 567, Valdivia 5090000, Chile; 4Laboratory of Cellular Communication, Program of Cell and Molecular Biology, Institute of Biomedical Sciences (ICBM), Faculty of Medicine, University of Chile, Av. Independencia 1027, Santiago 8380453, Chile; 5Instituto de Anatomía, Histología y Patología, Facultad de Medicina, Universidad Austral de Chile, Valdivia 5090000, Chile; 6Jeffrey Modell Center of Diagnosis and Research in Primary Immunodeficiencies. Faculty of Medicine, University of La Frontera, Temuco 4780000, Chile; 7NM MultiMat, Instituto de Ciencias Físicas y Matemáticas, Facultad de Ciencias, Universidad Austral de Chile, Valdivia 5090000, Chile

**Keywords:** self-assembly, sponges, ionic nanocomplexes, polyarginine, hyaluronic acid, cell proliferation

## Abstract

We report on the design, development, characterization, and a preliminary cellular evaluation of a novel solid material. This material is composed of low-molecular-weight hyaluronic acid (LMWHA) and polyarginine (PArg), which generate aqueous ionic nanocomplexes (INC) that are then freeze-dried to create the final product. Different ratios of LMWHA/PArg were selected to elaborate INC, the size and zeta potential of which ranged from 100 to 200 nm and +25 to −43 mV, respectively. Turbidimetry and nanoparticle concentration analyses demonstrated the high capacity of the INC to interact with increasing concentrations of LMWHA, improving the yield of production of the nanostructures. Interestingly, once the selected formulations of INC were freeze-dried, only those comprising a larger excess of LMWHA could form reproducible sponge formulations, as seen with the naked eye. This optical behavior was consistent with the scanning transmission electron microscopy (STEM) images, which showed a tendency of the particles to agglomerate when an excess of LMWHA was present. Mechanical characterization evidenced low stiffness in the materials, attributed to the low density and high porosity. A preliminary cellular evaluation in a fibroblast cell line (RMF-EG) evidenced the concentration range where swollen formulations did not affect cell proliferation (93–464 µM) at 24, 48, or 72 h. Considering that the reproducible sponge formulations were elaborated following inexpensive and non-contaminant methods and comprised bioactive components, we postulate them with potential for biomedical purposes. Additionally, this systematic study provides important information to design reproducible porous solid materials using ionic nanocomplexes.

## 1. Introduction

Aqueous ionic nanocomplexes (INC) are structures formed in water by the association of high-molecular-weight molecules, such as polymers, with complementary charged low- or high-molecular-weight polyions, such as ionic cross-linkers (tripolyphosphate), dyes, oligomers, or polyelectrolytes [1,2,3,4,5,6,7,8,9,10]. INC can be formed following the very simple procedure of mixing aqueous solutions of oppositely charged components at room temperature. This procedure avoids the use of organic/toxic solvents and the application of high mechanical energies, thus being ideal for biological uses. In addition, the procedure is low cost, both economically and environmentally, which facilitates the adoption of INC at the industrial level. The main nanostructures that are obtained as INC are nanogels, massive nanoprecipitates, or swollen aggregates [8,11,12,13,14]. Opposite to gelation (where only the larger component is able to be allocated in the surface, determining the surfacial charge), ionic complexation between two polymeric species allows the net charge of the INC to be selected by simply varying the ratio between oppositely charged species, thus allowing the production of formulations with tuneable electrostatic characteristics [13]. Depending on the binding forces between the ionic reactants and the ratio between their corresponding apparent charge concentrations, colloidal suspensions of the INC can be achieved just by tuning the absolute amount of matter in the mixture. At high electroneutralization regimes, the system tends to produce macroprecipitates, so the colloidal suspensions need to be highly diluted to keep colloidal stability [8]. However, an excess of one of the components allows stability of the colloidal suspension at a more concentrated regime, producing colloidal particles charged enough to ensure stability through electrostatic repulsions. Although the molecule to add in excess is normally of high molecular weight [8], examples are also found in which a low-molecular-weight component is added in excess, which, instead of diffusing out of the colloidal particles, keeps associated to them, determining the net charge of the particle and thus being responsible for the mixture stability [4].

Interestingly, upon removal of water from colloidal suspensions of INC, a solid material may arise [7,8]. In this sense, freeze-drying appears as a suitable technique to obtain micro- and nanoporous materials from suspensions of INC. Sponges are solid porous structures that can easily be manipulated and applied to selected biological tissues [15,16,17]. Polymeric materials prepared in a spongy form can be very useful in tissue engineering as scaffolds, which can reinforce, replace, and support some organs of the body, and also as non-scaffold materials that are able to promote cell growth [18]. If they are enriched with drugs, they are also usable as active drug delivery systems. In any case, such formulations must possess several essential properties, such as biocompatibility, biodegradability (if necessary), and absence of cytotoxicity, which primarily depend on the composition and the elaboration method of the material. Nowadays, different techniques have been developed to prepare sponge-like structures from polymers, such as phase separation, electrospinning, freeze-drying, etc.

Glycosaminoglycans, such as hyaluronic acid (HA), keratan sulfate, and chondroitin sulfate, are negatively charged biopolymers and good candidates to provide biological and functional properties as they are part of the extracellular matrix in a variety of tissues. HA is composed of repeating units of disaccharides, which include D-glucuronic acid and N-acetylglucosamine molecules linked by –(1–4) and –(1–3) glycoside bonds. This compound is involved in numerous processes occurring in the body, such as wound healing, ovulation, fertilization, signal transduction, and tumor physiology [19]. The biocompatibility and related negligible side effects make HA one of the more readily available compounds used in many fields of medicine as a biologically active molecule and as excipient in drug delivery systems [19,20].

Polyaminoacids are promising macromolecules for the development of biological active compounds and drug delivery systems. This is based on the fact that these molecules are structurally similar to polypeptides and are thus degraded by human enzymes; their accumulation within the organism is minimal. Interestingly, the cationic polyaminoacid polyarginine (PArg) shows interesting biological properties, such as being able to translocate through cell membranes, thereby promoting the uptake of molecules associated with it [21,22]. This interesting feature of PArg has been exploited to develop drug delivery systems used for gene therapy [23], protein/vaccine delivery [24], and cancer treatment [25]. Furthermore, PArg enhances the absorption of drugs across epithelia [26], a property that may be utilized for mucosal drug delivery. Interestingly, the toxicity of the bare polycationic PArg may be minimized by its complexation with polyanionic species, such as HA [27].

The aim of this work was to study, for the first time, the formation of reproducible sponges using INC comprising LMWHA and PArg prepared at different LMWHA/PArg as the input. The methodology to prepare the INC, which was further used to test sponge formation, was similar to the one previously developed by us [13] but explored new combinations. The colloidal suspensions were studied in terms of turbidity, nanoparticle concentration, apparent hydrodynamic diameter, zeta potential, and shape. The solid materials were prepared by freeze-drying (using standard conditions: 0.02 mbar, −54 °C, and 24 h) the obtained INC suspensions, a strategy that is significantly different from others focused at obtaining solid materials from solubilized components [28,29,30,31]. Optical and scanning electronic microscopy (SEM) images of the solid material (sponge) were used to support the analysis of the final product. Mechanical characterization evidenced low stiffness in the materials, attributed to the low density and high porosity. Finally, we conducted a preliminary cellular evaluation in fibroblast (RMF-EG cell line). This evidenced the concentration range where swollen formulations did not affect cell proliferation at 24, 48, or 72 h, thus projecting potential doses to be administered in further in vitro or preclinical/clinical studies.

## 2. Materials and Methods

### 2.1. Chemicals

Low-molecular-weight hyaluronic acid (LMWHA, Mw ~29 kDa) was purchased from Inquiaroma (Barcelona, Spain). The equivalent weight of the LMWHA considering the number of possible charges was 403 g/mol of ionizable groups. Polyarginine (PArg, Mw ~5–15 kDa) was purchased from Sigma Aldrich (St. Louis, MO, USA). The equivalent weight of the PArg considering the number of possible charges was 192.5 g/mol of ionizable groups. For cell culture studies (RMF-EG cells), we used Dulbecco’s Modified Eagle Medium (DMEM-HG, Gibco, Paisley, UK), fetal bovine serum (FBS, Gibco, Paisley, UK), and 1% penicillin–streptomycin (Gibco, Paisley, UK). 3-(4,5-dimethylthiazol-2-yl)-5-(3-carboxymethoxyphenyl)-2-(4-sulfophen-yl)-2H-tetrazolium inner salt (MTS) and phenazine methosulfate (PMS) were purchased from Promega (Madison, WI, USA). All other reagents were of the highest analytical grade. Milli-Q water was used for experimentation.

### 2.2. Solid Material Preparation and Characterization

The method comprised three steps: (1) prepare suspensions of polymeric INC containing LMWHA and PArg; (2) freeze-dry the obtained INC suspensions to form the solid material; and (3) sterilize the solid material resulting from freeze-drying the INC suspensions.

INC suspensions were prepared following a procedure similar to the one described by Oyarzun-Ampuero et al. [13]. Different mass of LMWHA (5, 10, 12, 13, 15, 20, 25, 30, 35, and 50 µmol) were dissolved in 4.5 mL of Milli-Q water and added to a solution prepared by dissolving 12 µmol of PArg in 4.5 mL of Milli-Q water. The mixing was done in 50-mL cylindrical plastic containers with 38 mm diameter while stirring at room temperature. Magnetic stirring was maintained for 10 min to enable complete stabilization of the systems. The formed INC suspensions were then characterized. Size and zeta potential were determined by photon correlation spectroscopy and laser Doppler anemometry using a Zetasizer Nano-ZS (Malvern Instruments, Malvern, UK). Each batch was analyzed in triplicate. Turbidimetry studies were done using a UV–Vis spectrophotometer (Perkin Elmer, Lambda 25, Waltham, MA, USA), choosing a wavelength where the individual components (LMWHA and PArg) did not present absorption bands. In this research, the wavelength of 540 nm was selected. The nanoparticle concentration was determined by nanoparticle tracking analysis (NTA) using a NanoSight NS300 (Malvern Instruments, Malvern, UK). Each batch was diluted from 10 to 1000 times with Milli-Q water to achieve an optimum concentration range of 10^7^–10^9^ particles/mL. A minimum of five videos (one minute each one) of the particles moving under Brownian motion were captured by the NanoSight. The videos were then analyzed for size distribution and particle concentration using the built-in NTA v 3.0 software (Malvern, UK). The morphology of the INC suspensions was determined by scanning transmission electron microscopy (STEM), model Inspect F-50 (FEI, Hillsboro, OR, USA). STEM images were obtained by sticking a droplet (20 µL) of the formulation to a cooper grid (200 mesh, covered with Formvar) for 2 min, then removing the droplet with filter paper (avoiding the paper from touching the grid), then washing the grid twice with a droplet of Milli-Q water for 1 min, and then removing the droplet with filter paper. Subsequently, the sample was stained with a solution of 1% (w/v) phosphotungstic acid by adding a droplet of this solution to the grid for 2 min and then removing with filter paper. Finally, the grid was dried at room temperature for at least 1 h before being analyzed.

In order to prepare the solid materials, 9 mL of the prepared INC suspensions were frozen at −20 °C for 24 h in the cylindrical plastic container they were produced in and then transferred to a freeze-dryer (Christ, Alpha Plus 1-2 LD, Osterode am Harz, Germany). The sublimation proceeded at 0.02 mbar for 24 h (condenser temperature of −54 °C). The morphology and porosity of the solid materials were examined with naked eye and by SEM (LEO 420, Cambrigde, England). For SEM analysis, the samples were cut with a razor blade and coated with a gold layer. Porosity threshold of the sponges was analyzed theoretically after their non-floatability in cyclohexane, a low-density organic solvent, was corroborated so that the maximum volume limit value for the solid part of the sponges could be easily calculated from their mass. The porosity threshold was then calculated as follows:
$$Porosity=\frac{volume\;of\;the\;sponges-maximum\;volume\;limit\;of\;the\;solid\;part\;of\;the\;sponges}{volume\;of\;the\;sponges}$$

For mechanical characterizations of the sponges, a series of 200-µL samples were transferred in 96-well plates, frozen (−20 °C, 24 h), and then freeze-dried (using standard conditions: 0.02 mbar, −54 °C, and 24 h). The analyses were performed in the compression mode, and hardness and apparent Young’s modulus (E_app_) were obtained. The materials were placed on a fixture base table (TA-BTKIT, Brookfield) and compressed, carefully centered, with a cylindrical TA-39 probe of 2 mm diameter. The resolution of the texture analysis system was 0.1 g and 0.1 mm. The test speed was set at 0.7 mm s^−1^, the load trigger value ranged from 0.7 to 1.0 g, and the maximum load was set at 2 g. For the final E_app_ analysis, data for 25 sponges were considered. Finally, the stability of the sponges in water and PBS was evaluated by optical microscopy (Olympus CKX41, Arquimed, Tokyo, Japan) using a digital camera (Digital Sight DS-Fi2, Nikon, Tokyo, Japan) with the Micrometrics SE Premium^®^ software. The samples were placed at 37 °C (room temperature) on a slide and under the 4× objective. Subsequently, 10 µL of Milli-Q water or PBS was added, and the behavior of each sponge was recorded using the Open Broadcaster software (v.23.0.2, OBS Studios Contributors). To sterilize the solid material resulting from freeze-drying the INC suspensions, the sponges were sterilized under ultraviolet 25 W light for 4 min (UV lamp, Biolight, Santiago, Chile), sealed in plastic bags in laminar flow hood, and stored in desiccators containing dried silica gel in order to avoid moisture before their use. This method was previously used by our group, and the resulting sponges were demonstrated to maintain their biological potential in in vitro [32,33] and in vivo [34] studies.

### 2.3. Cellular Studies

#### 2.3.1. Material Preparation and Administration to Cells

A formulation of INC comprising 12 mg of LMWHA and 2.4 mg and PArg (charge ratio LMWHA/PArg = 2.4), was prepared following the procedure described in Section 2.2. Aliquots of the above preparation were diluted in Milli-Q water in order to obtain 93, 186, 464, and 1856 µM in 100 µL; transferred in 96-well plates; and freeze-dried as described in Section 2.2. For the transfer of the material to the cells, 100 µL of culture medium was added to the sponges in the plates, and swollen formulations were then pipetted to the 96-well plates containing the cells.

#### 2.3.2. Proliferation Assay

Five thousand RMF-EG cells were seeded in 96-well plates. After the cells adhered to the plates (2 h), the culture medium (100 µL) was extracted, and the selected formulation (charge ratio LMWHA/PArg = 2.4, prepared and transferred as described in Section 2.3.1) was added in order to achieve different concentrations (93, 185, 464, and 1856 µM). After the evaluation in terms of cellular proliferation at different time intervals (24, 48, and 72 h), the medium was replaced with 80 µL of serum-free medium plus 20 µL of MTS:PMS (20:1), mixed, and then incubated for 2 h at 37 °C. We used proliferation kit MTS (Promega, Madison, WI, USA). The reduction of MTS to formazan was determined by measuring the absorbance of this solution at 490 nm by spectrophotometry. This methodology is in accordance with the ISO 10993-5 guidelines.

### 2.4. Statistical Analysis

The results are shown as the mean ± standard error of the mean for *n* = 3. The results were analyzed using one-way ANOVA tests and Dunn posttests. Statistical significance was set at *p* < 0.05.

## 3. Results and Discussion

### 3.1. LMWHA/PArg INC Suspensions

Table 1 shows the results of apparent size and zeta potential of the prepared LMWHA/PArg INC. The methodology followed involved fixing the amount of PArg and varying the amount of LMWHA in order to achieve different charge ratios. It can be seen that most of the mixtures showed apparent size in the range between 100 and 200 nm, with low polydispersity indexes. Interestingly, macroprecipitates were formed at a LMWHA/PArg ratio of 1.1, expressed in relative number of equivalents, evidencing the highest electroneutralization between negatively and positively charged polymers. In this respect, it can also be seen that, at lower ratios, stable INC with positive zeta potential were obtained due to the excess of the positively charged PArg, while at higher ratios, the INC showed negative zeta potential due to the excess of LMWHA in the particles. In addition, there were appreciable differences in size, zeta potential, and polydispersity. These characteristics were influenced by the specific properties of each component (i.e., rigidity, linear charge, and molecular weight) and by the total mass of the polymeric formulations. In fact, formulations developed under the same strategy but using high-molecular-weight polymers showed higher size and higher polydispersity [13,35,36], indicating the role of the polymeric molecular weight on the homogeneity of the mixtures. The high correlation between the charge ratio of the components and the physicochemical properties of the formulations could be attributed to the low molecular weight of the polymers and also to their low molecular mass polydispersion. These selected parameters (concentration, charge ratio, and molecular weight of polymers) are ideal for designing specific nanoformulations in terms of size, low size polydispersity, and net charge.

With the aim of more in-depth characterization of the colloidal suspensions, turbidity and nanoparticle tracking analyses of the formulations were studied. Due to the fact that stable colloidal suspensions maintain INC homogeneously dispersed in the aqueous phase, turbidity may give information about the stability of the suspensions as well as a qualitative idea regarding the interplay of size and amount of INC formed [37]. Figure 1 shows the values of the apparent absorbance of the colloidal suspensions at 540 nm, where functional groups of LMWHA and PArg did not show absorption bands. It can be seen that there was almost a linear increase in turbidity as the total mass of LMWHA increased (not considering the precipitation zone) due to an increase in the mass of the suspensions. The increase in the size of the INC as more LMWHA was added was moderate, as can be seen in Table 1. Therefore, the increase in turbidity was presumably mainly caused by the increase in the number of formed nanoparticles.

The above presumption was corroborated by nanoparticle tracking analysis, which showed that the concentration of nanoparticles in the colloidal suspensions increased similarly to the turbidity. In fact, a linear tendency related to the increases in LMWHA mass and LMWHA/PArg ratio was also observed. Because the amount of PArg was fixed in these experiments, the increase in the number and size of particles indicate that the formation mechanism of the INC allowed the incorporation of more LMWHA to preexisting particles, making them generally bigger and showing higher zeta potential (in absolute value), and that the low-molecular-weight polymers were subjected to interaction equilibrium, allowing a higher number of contacts as the absolute concentration of the reactants increased. Importantly, as evidenced above, turbidity (analyzed by UV–Vis spectrophotometer) represents a simple and inexpensive methodology to preliminarily study (in terms of colloidal behavior) nanoparticle concentration and stability to design new nanoparticle formulations. It could also be useful to analyze batch-to-batch reproducibility for routinary analyses.

STEM microscopy experiments were developed in order to visualize selected formulations (LMWHA/PArg: 0.4, 0.8, 1.2, 1.6, 2.0, 2.4). As evidenced in Figure 2, the nanoparticles showed a spheroidal shape with an apparent size between 100 and 200 nm. In addition, an aggregation/agglomeration pattern could be seen as the LMWHA/PArg ratio increased. The samples at charge ratio of ≤1.6 showed more homogenous distribution with lower aggregation between the particles, while an agglomeration pattern between the nanoparticles was observed at higher ratios. It can be expected that the observed agglomeration pattern at higher mass of LMWHA could influence the formation of the solid material from the LMWHA/PArg nanocomplexes.

### 3.2. Solid Materials

Figure 3 shows the obtained materials after freeze-drying 9 mL of the colloidal suspensions, whose characteristics are shown in Table 1 and Figure 1 and Figure 2, in 50-mL cylindrical containers with 38 mm diameter. The selected LMWHA/PArg ratios were 0.4, 0.8, 1.2, 1.6, 2.0, and 2.4. It can be noticed that the formation of well-structured sponges failed in the case of the two compositions bearing excess of PArg as well as when freeze-drying the highly-neutralized mixture showing a LMWHA/PArg ratio of 1.2. In contrast, the suspensions showing an excess of LMWHA presented better characteristics as sponges, showing quasi cylindrical shape with diameter of around 30 mm, which was slightly lower than that of the container they were produced in, and height of 3–6 mm, as a result of shrinking on their z axis. The formed sponges were low-density, highly porous materials with density lower than 10^−3^ g/cm^3^ and porosity higher than 99%. Interestingly, we observed by compression analyses that the sponges had low stiffness. Their stress–strain curves showed a nonlinear behavior with viscoelastic characteristics (Appendix A). Hardness of around 8 g was obtained between 7–15% deformation, and the E_app_ values were around 253 ± 77 kPa (n = 25), a fact related to the low density and high porosity of the materials. In addition, water and PBS were added to the sponges at 37 °C, and the behavior was observed by microscopy. As shown in the videos (video S1, video S2), the microfibers were rapidly hydrated, and the material were dispersed in small pieces into the medium. Materials with similar behavior when exposed to biorelevant media have been proposed for in vitro/in vivo testing for wound healing purposes [32,34,38] and/or to be enriched with active molecules for therapeutic purposes [39,40,41]. Furthermore, other interesting characterizations regarding the interplay between cells and solid materials and obtaining the stiffness have been published. In this sense, the proposal from Liverani et. al. (2017) could represent an approach to be considered in the future to characterize this and other solid materials [42].

The production of well-formed sponges from a colloidal suspension depends on several factors, among which we can name the shape and dimensions of the container compared to the volume of the colloidal suspension, the amount of components of the colloidal suspensions, and the nature of the material components. Upon freezing colloidal suspensions, the colloidal particles are concentrated during ice formation at the boundary of ice crystals, submitted to the out-of-equilibrium process called “ice-segregation-induced self-assembly” (ISISA) [8,43,44]. The migration of the solutes during freezing is determined by their hydrophilicity and mutual interactions at increasing local concentrations and lower temperatures. In this sense, as freezing at temperatures around −20 °C is produced from outside to inside the limits of the suspension volume, small ions may migrate with liquid water and concentrate at the inner part of the frozen cylinder. Molecules and particles with lower diffusion coefficients, interacting with complementary charged polyions, and amphiphilic or hydrophobic components may migrate more slowly, producing structured deposits at the boundary of ice crystals. Molecular rigidity may enhance the cohesive forces between particles and molecules subjected to electrostatic interactions. Sublimation of ice crystals after freezing during freeze-drying once the solutes are rearranged at the boundary of ice crystals then allows well-structured porous materials to be obtained. In this sense, several facts may explain the results found in this investigation. PArg is a more flexible, more amphiphilic polymer compared with the more rigid, more hydrophilic HA. In addition, the compositions with a lower content of LMWHA present less total mass of solutes. All this favors migration of the components, a weak structure at the boundary of ice crystals during freezing, and a tendency of the system to collapse during sublimation due to attractive interactions and gravity. In contrast, the higher total mass of the LMWHA-rich compositions and molecular rigidity of the polysaccharide favor the reinforcement of the structures around ice crystals and achieve the necessary tensile strength to keep the porous structure during and after sublimation.

The obtained materials presented a porous structure made of a combination of micrometric morphologies, such as microfibers and microsheets, showing a high surface area, as can be seen from the SEM images in Figure 4. As evidenced, materials with LMWHA/PArg ratios of ≤1.2 showed a more entangled and holey network of microstructures and lower pore size. As the LMWHA content increased in the materials, higher pores were observed, which was related to the formation of more extended, non-collapsed microstructures, in accordance with the facts revealed by optical observation. In addition, it can be seen that, at ratios of ≥1.6, larger fibers and microsheets were increasingly formed, which is in agreement with the tendency described for the nanoparticles evidenced by STEM (Figure 2) and for the sponges evidenced by optical images showing more structured materials (Figure 3). Interestingly, the images did not show texturing on the surface of the laminar structures. This fact suggests that there were no remaining molecules (such as uncomplexed polymers). The above can be explained by the tendency of the solutes to migrate in the freezing process, i.e., once ice is produced upon freezing, uncomplexed chains (if any) will migrate and bind the aggregates as the local concentration increases, making them part of the final product.

### 3.3. Cellular Studies

Fibroblasts represent an adequate model cell to test the safety of new materials proposed to be applied in the skin for different therapeutic purposes [45,46,47]. Sponges based on a LMWHA/PArg ratio of 2.4 were selected due to their more structured characteristics (Figure 3 and Figure 4) and also because they show very fast swellability when exposed to aqueous media. The abovementioned characteristics are important because they favor a better manipulation of the solid formulations to be applied or when swollen formulations are administered (as in the present study). Different doses of the sponges treated with culture medium were transferred to cells in order to achieve concentrations of 93, 185, 464, and 1856 µM (see Section 2.3). As evidenced in Figure 5, the concentrations of 93, 185, and 464 µM did not significantly affect the fibroblast cell proliferation at 24, 48, or 72 h. The absence of toxicity for those doses could be used as a reference for further in vitro or preclinical/clinical studies. In contrast, the highest concentration (1856 µM) decreased the cell proliferation from 48 h. Additionally, cell morphology could be clearly seen in phase-contrast microphotographs (Appendix A). The results were similar to those observed by MTS, demonstrating that a considerable amount of dead fibroblast was observed at the highest concentration (1586 µM). In fact, at 932 µM, cells started to accumulate, showing a less healthy appearance. This behavior could be reasonably explained due to the larger content of polymeric species in the culture, possibly affecting the homeostatic equilibrium between the cells, the culture medium, and the transfer of O_2_/CO_2_ to the environment.

### 3.4. Final Remarks

Low-density sponge-like materials, as those presented in this paper, are solid pharmaceutical forms to be administered in tissues and can easily be obtained from INC. The formation of INC between biocompatible ionic polymers therefore emerges as a useful tool to achieve solid, highly porous materials with biomedical potential. Thus, the strategy for this type of formulations is (i) to formulate INC that provides good balance of charges and intimate mixing of oppositely charged polymeric species and (ii) to fabricate low-density sponges by freeze-drying (avoiding cryoprotectants). This has several advantages for better manipulation, such as high stability, easy storage and transportation [38], together with high therapeutic potential [7,8,32,34,38,39,41] despite their low stiffness and high lability toward hydration (see video S1, video S2). Importantly, the hydrated material stands as a gel-like mucus (if intending to dry INC by freeze-drying and further reconstitute them through hydration, cryoprotectants are required [36,48,49]), and cell cultures studies (done following the ISO 10993-5 guidelines) correspond to the cellular response to this hydrated hydrogel. The physical and mechanical characteristics of the sponges facilitate their administration to patients as they are able to be directly applied into selected tissues [34,50,51,52,53,54]. Due to the need to keep stability of the colloidal suspensions of INC to be freeze-dried, the total maximum concentration of reactants is normally low, furnishing low density to the final solid materials and avoiding excessive metabolic stress when applied, which is adequate for therapeutic purposes. Another advantage of these materials regarding possible commercial purposes is that they are elaborated under mild conditions in aqueous medium, avoiding toxic excipients or covalent cross-linkers. In addition, these materials can also serve as drug carriers due to the countless number of active molecules that can be incorporated in the colloidal suspensions, giving rise to the solid materials. The incorporation of the extra active molecules must be studied case-by-case for their possible interactions with the polymeric reactants, their influence on INC formation and stability [7,41], and their behavior in the freeze-drying process.

Here, we have shown the formation of low-density sponge-like materials made from INC of low-molecular-weight polymers, among which one was an anionic polysaccharide (LMWHA) and the other was a cationic polyaminoacid (PArg). Both higher stability as a solid material and good swellability in culture media were furnished by an excess of LMWHA over PArg (LMWHA/PArg ratio of 2.4). In vitro experiments showed a limit of cytotoxic concentration of this selected sponge for RMF-EG fibroblasts in the range of 464–1856 µM. This affords potential safe doses for further studies on the application of this material for medical purposes. The critical role of PArg in these materials as a biodegradable and biocompatible polycationic substrate able to interact with LMWHA, allowing the formation of the corresponding INC, could also be further extended to specific uses where its cell-penetrating capacities could promote the intracellular access of selected drugs and genes attached to the solid material [55,56,57].

## 4. Conclusions

In the present work, we demonstrated, for the first time, that aqueous INC formed by LMWHA and PArg were able to generate sponges after freeze-drying of the nanosuspensions. Interestingly, only those INC comprising an excess of HA were able to form sponges. NTA showed the formation of an increasing number of INC as the excess of LMWHA increased up to 2.4 over PArg. STEM experiments showed an increasing tendency of the particles to agglomerate. This phenomenon may be attributable to the higher total mass of these formulations, together with the higher rigidity afforded by HA. Mechanical characterization evidenced materials with low stiffness, attributed to the low density and high porosity. Finally, we provided a preliminary cellular evaluation in fibroblasts, evidencing the concentration range where a selected formulation did not affect cell proliferation up to 72 h, thus projecting potential doses to be administered in further in vitro or preclinical/clinical studies. Considering that the generated materials are composed of biodegradable and biocompatible compounds, we postulate them as candidates with potential for biomedical purposes. Additionally, this systematic study provides important information for researchers to design reproducible porous solid materials using INC of selected compositions as input.

## Figures and Tables

**Figure 1 nanomaterials-09-00944-f001:**
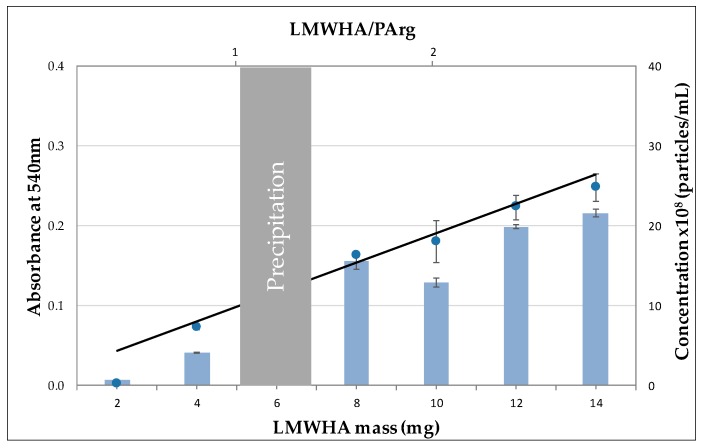
Turbidity (dots) and nanoparticle concentration (bars) as a function of the LMWHA mass and ratio of the components (LMWHA/PArg) (mean ± SD, *n* = 3).

**Figure 2 nanomaterials-09-00944-f002:**
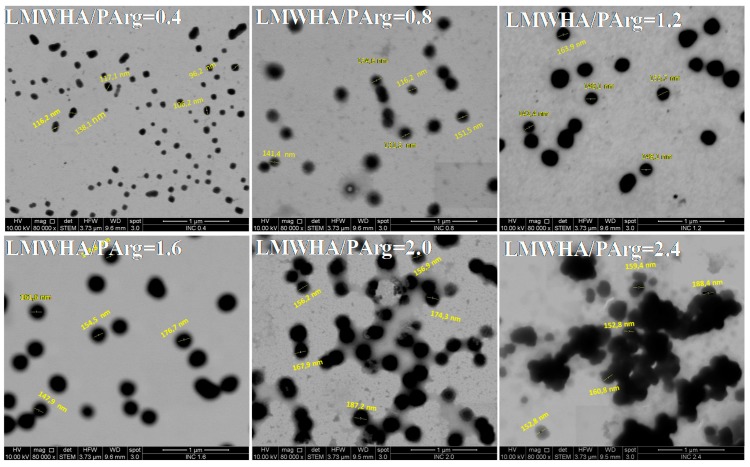
Scanning transmission electron microscopy (STEM) images of ionic nanocomplexes (INC) containing LMWHA and PArg at a ratio of 0.4, 0.8, 1.2, 1.6, 2.0, and 2.4 (scale bar of 1 µm).

**Figure 3 nanomaterials-09-00944-f003:**
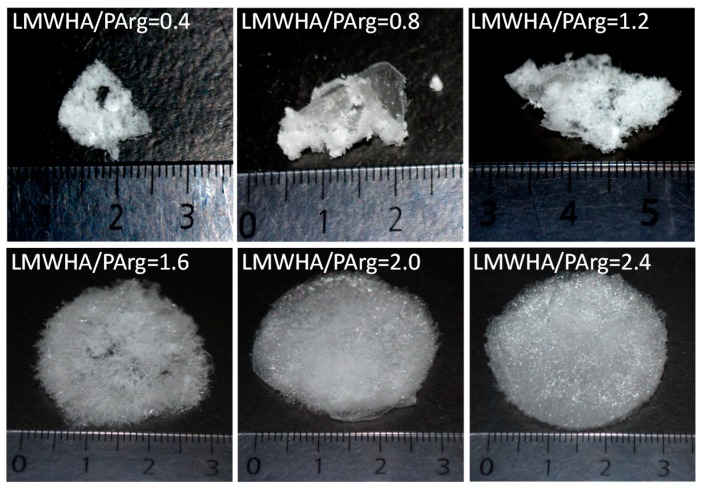
Optical images of materials obtained after freeze-drying 9 mL of INC aqueous suspensions containing LMWHA and PArg at a ratio of 0.4, 0.8, 1.2, 1.6, 2.0, and 2.4 in cylindrical plastic containers with 38 mm diameter.

**Figure 4 nanomaterials-09-00944-f004:**
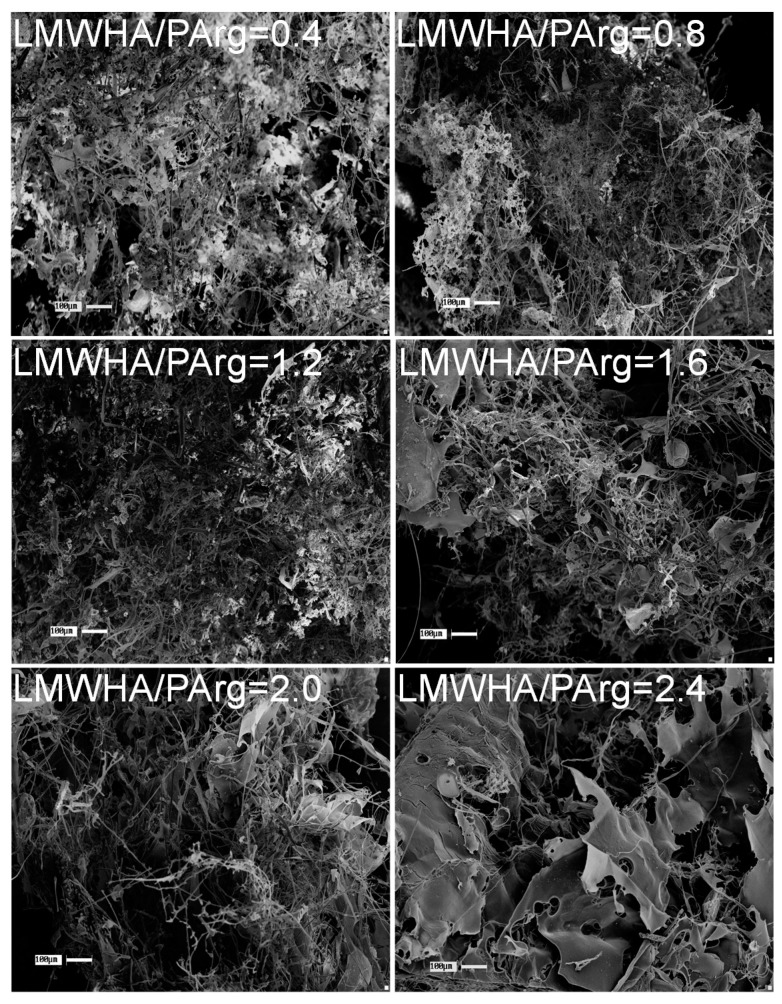
SEM images of materials obtained after freeze-drying 9 mL of INC containing LMWHA and PArg at different ratios of 0.4, 0.8, 1.2, 1.6, 2.0, and 2.4 in cylindrical plastic containers with 38 mm diameter. Scale bar of 100 mm.

**Figure 5 nanomaterials-09-00944-f005:**
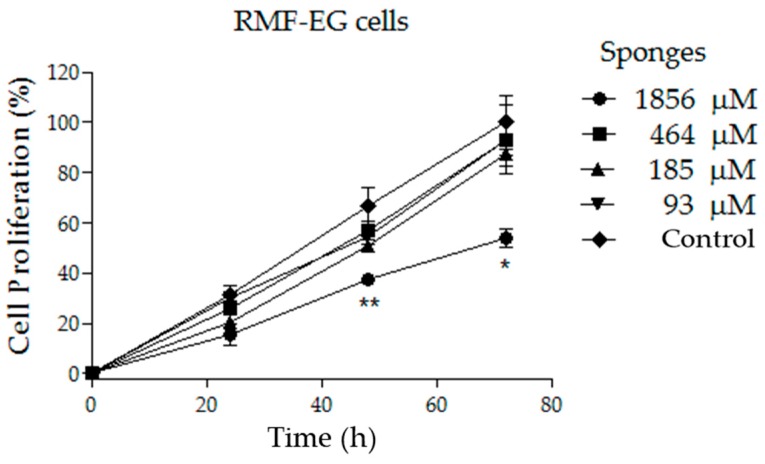
Proliferation curve of fibroblast (RMF-EG cell line) exposed to different doses of sponges of LMWHA/PArg = 2.4. Significant differences were obtained when comparing the values from each time with respect to the respective control value (mean ± SD, *n* = 3) and are indicated (** *p* ≤ 0.01, * *p* ≤ 0.05).

**Table 1 nanomaterials-09-00944-t001:** Physicochemical properties of formulations prepared with different ratios of low-molecular-weight hyaluronic acid/polyarginine (LMWHA/PArg) and evaluated in Milli-Q water (mean ± SD, *n* = 3).

Mass Ratio LMWHA/PArg	Charge Ratio [LMWHA]/[PArg]	Size (nm)	Polydispersity Index	Zeta Potential (mV)
2.0/2.4	0.4	126 ± 29	0.3–0.4	21.6 ± 5
4.0/2.4	0.8	128 ± 2	0.1–0.2	24.35 ± 3
4.7/2.4	0.9	138 ± 23	0.1–0.2	23.3 ± 3
5.4/2.4	1.1	Precipitation	---	---
6.0/2.4	1.2	141 ± 10	0.1–0.2	−17.9 ± 3
8.0/2.4	1.6	145 ± 9	0.2–0.3	−33.1 ± 3
10/2.4	2.0	149 ± 32	0.2–0.3	−35.0 ± 4
12/2.4	2.4	146 ± 18	0.2–0.3	−36.2 ± 3
14/2.4	2.8	186 ± 71	0.1–0.2	−39.7 ± 0.4
20/2.4	4.0	166 ± 13	0.1–0.2	−42.6 ± 0.1

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
