# Peer review of "Ionic Nanocomplexes of Hyaluronic Acid and Polyarginine to Form Solid Materials: A Green Methodology to Obtain Sponges with Biomedical Potential"

_nanomaterials, 2019, doi:10.3390/nano9070944_

Reviewer 1 Report

The paper can be acepted in the present form. The authors strongly improved the quality of the manuscript by adding also some more analyses.

- Figure S1 is unfortunately missing. Please, add into the supplemntary information.

Author Response

The responses have been uploaded.

Reviewer 2 Report

The authors Villamizar-Sarmineto et al report the development of a innovative 3D scaffold based on ionic nanocomplexes of hyaluronic acid and polyarginine for biomedical application. The authors characterized the physiochemical proprieties the nanocomplex suspension in terms of size, PDI, zeta potential and turbidity. The obtained innovative solid material was characterized through STEM and SEM analysis. Finally preliminary results on cellular proliferation was provided using fibroblast cell line.

The paper introduce an innovative device as candidate for potential biomedical purposes. The manuscript looks like well written and organized, however a closer look at the draft shows that some additional analysis are required.

1.       SEM images of fibrobasts cultured within the scaffold should be performed

2.       ematoxilin and eosin images of fibrobasts cultured within the scaffold should be performed

3.       Some prototype to test the mechanical proprieties of natural and synthetic devices have been already developed, an example is the work of Ibrahim group Investigating the Mechanobiology of Cancer Cell–ECM Interaction Through Collagen-Based 3D Scaffolds which should be included in the manuscript.

The paper should be considered after major revisions.

 Author Response

The responses have been uploaded.

Reviewer 3 Report

This work deals with the preparation and characterization of ionic nanocomplexes of hyaluronic acid (HA) and polyarginine(PArg)and with that of their solid compounds obtained after freeze-drying.

Abstract: In the first line authors claim that they investigate a novel solid material, but more information should be given with this respect (type of compound, polymers). PArg and HA are treated as “excipients”, which is not in accordance with the proposed biomedical purposes. Line 36-37 is not clear, “which agrees with the behavior observed for INC in electronic miscorcopy” ..but this behaviour is not explained.

 Introduction:

-The differences between ionotropic gelation and ionic complexation are not clear.

-If the ionic nanogels were previously reported. What is the novelty of this work? It should be highlighted.

 Materials and methods:

- The equivalent moles should be employed, and not the equivalent weights.

-Sterilization of polysaccharides is a crucial point in their further application. Prepared materials were sterilized, but the effect of this sterilization on their degradation or molecular weight is not addressed. Some measurements are needed in this point.

-Samples are not purified. This is a weak point in the preparation of the samples. How do you eliminate uncomplexed polymers, because they will have a clear effect on the heterogeneity of the final samples (DLS, zeta potential.)

_Line 162-163. How the porosity was calculated should be expressed by an equation.

 Results:

-          Table 1: The observed effect should be the same by excess of positive polymer and by excess of negative polymer, however there are differences in the changes of size and zeta potential (absolute value). Why?. Clarify this point, please.

-          Line 225-226: Why turbidity increased as the mass of LMWHA increased?, why the dispersion of nanocomplexes are turbid? Turbidity doesnot correspond with particle sizes of 100-200 nm (Table 1)?. Results and discussion is not coherently addressed with this regard.

-          Figure 1. Transmittance is more appropriate than absorbance. Nevertheless the graph is easy to understand.

-          Line 237 and 238:”making them bigger and showing higher zeta potential..” But this is not observed in all the cases?¿.

-          Taking into account the nanometric size of nanocomplexes, they are not appropriate materials for wound healing purposes. Despite the dried solids are sponges-like structures, when they are put in contact with the physiological medium, they are dispersed again, they are not macroscopic hydrogels. This is a great incoherence in the work that should be explained to understand the proposed applicability of the nanocomplexes.

-          According to this, how did you carry out cellular studies if sponges are dispersed in the aqueous medium?

-          Line 282. What is the reactant and what the solute? This nomenclature seems to be out of order.

 Final remarks:

They are similar to conclusio

Author Response

The responses have been uploaded.

Round  2

Reviewer 2 Report

The work has been improved. The authors have addressed some issues.

As previously reported the mechanical characterization of the material is crucial in order to achieve biomedical application. In this regard, the use of innovative prototype able to specifically measure the stiffness of biomaterials represent an hot topic. For the above reasons the authors should include in the references the work titled Investigating the Mechanobiology of Cancer Cell–ECM Interaction Through Collagen-Based 3D Scaffolds.

 Minor revisions are requested before publication

Author Response

The response has been uploaded.

This manuscript is a resubmission of an earlier submission. The following is a list of the peer review reports and author responses from that submission.

Round  1

Reviewer 1 Report

Felipe and co-workers reported the development of sponge-like materials from the nano-scaled ionic complexes (INC) of low molecular weight hyaluronic acid (LMWHA) and polyarginine by freezing dying methods. STEM observation of the INC showed that spherical particles with 100 ~ 200 nm diameters were obtained in the presence of excess LMWHA and freezing dying of the dispersions of the spherical nanoparticles led to the formation of sponge-like materials with well-defined porous structures. The authors performed proliferation assay of RMF-EG cells in the presence on different amounts of the sponge-like materials and negligible cytotoxicity was observed up to the presence of 464 ~ 1856 uM of the materials.

Solid materials with porous materials are promising materials for tissue-engineering applications. The presented method based on the combination of polyionic complexation and freezing drying is simple but effective for the preparation of porous materials. Indeed, this method was able to be used for the preparation of porous materials employing biocompatible hyaluronic acid and polyarginine. The low cytotoxicity of the resulting porous materials evinced the potential use for the bio-medical application although the observed decrease of cell proliferation in the presence of the materials at high concentration conditions. 

Reviewer 2 Report

The manuscript entitled “Ionic nanocomplexes of hyaluronic acid and polyarginine to form solid materials: A green methodology to obtain sponges with biomedical potential” submitted by Maria Gabriela Villamizar-Sarmiento et al. and with manuscript ID: nanomaterials-457938, seems not to add any remarkable novelty in the field of nanomaterials in comparison with a previous publication from the corresponding author of the submitted article (A new drug nanocarrier consisting of polyarginine and hyaluronic acid by Felipe A. Oyarzun-Ampuero  et al. Eur. J. of Pharmac. Biopharmac. 79 (2011) 54-57).

 The green methodology claimed as a novelty in the submitted article to Nanomaterials journal does not have originality as is a method previously used to obtain sponges by other authors (Development of a freeze-dried skin care product composed of hyaluronic acid and poly(γ-glutamic acid) containing bioactive components for application after chemical peels by Yuka Isago et al. Open Journal of Regenerative Medicine, 2014,3, 43-53).

Reviewer 3 Report

The paper is focused on the design of ionic nanocomplexes sponges based on hyaluronic acid and polyarginine for biomedical applications. The paper is interesting, the introduction is well written, and also supported by several recently published papers on the topic. Nevertheless some mayor changes and some more analyses should be addressed by the authors in order to improve the clarity and the understanding and  support the conclusions.

- The authors should strongly improve the English.

- On page 3, line 119-121. The sentence is too long and not clear.

- Did the authors evaluate mechanical properties of the sponges? It would be interesting to understand the mechanical behaviour of these materials, especially if the authors address them for tissue engineering applications (i.e. skin).

- Did the authors evaluate the stability over time of the LMWHA/PArg sponges in water at 37°C? Did they assess the swelling behaviour of the materials? It could be interesting to correlate this behaviour to cell/material interaction.

- Did the authors optimize the freeze drying process in terms of condenser temperature, time, and pressure? There is a strict correlation between these factors which can influence the scaffold realization.

- The authors performed some indirect cytotoxicity tests, it would be nice to study the direct cell/material interaction by seeding cells on top of the scaffolds. In such way, the authors could also understand the role of PArg on cell adhesion/proliferation.Furthemore, did the authors follow ISO 10993-5 guidelines?